# Coralgal reef morphology records punctuated sea-level rise during the last deglaciation

Pankaj Khanna [1], André W. Droxler[1], Jeffrey A. Nittrouer[1], John W. Tunnell Jr[2] & Thomas C. Shirley[3]

Coralgal reefs preserve the signatures of sea-level fluctuations over Earth's history, in particular since the Last Glacial Maximum 20,000 years ago, and are used in this study to indicate that punctuated sea-level rise events are more common than previously observed during the last deglaciation. Recognizing the nature of past sea-level rises (i.e., gradual or stepwise) during deglaciation is critical for informing models that predict future vertical behavior of global oceans. Here we present high-resolution bathymetric and seismic sonar data sets of 10 morphologically similar drowned reefs that grew during the last deglaciation and spread 120 km apart along the south Texas shelf edge. Herein, six commonly observed terrace levels are interpreted to be generated by several punctuated sea-level rise events forcing the reefs to shrink and backstep through time. These systematic and common terraces are interpreted to record punctuated sea-level rise events over timescales of decades to centuries during the last deglaciation, previously recognized only during the late Holocene.

[1] Department of Earth, Environmental and Planetary Sciences, Rice University, 6100 Main St, Houston, TX 77005, USA. [2] Harte Research Institute for Gulf of Mexico Studies TAMU-CC, 6300 Ocean Dr., Corpus Christi, TX 78412, USA. [3] Department of Life Sciences, TAMU-CC, 6300 Ocean Dr., Corpus Christi, TX 78412, USA. Correspondence and requests for materials should be addressed to P.K. (email: pankaj89@gmail.com) or to A.W.D. (email: andre@rice.edu)

Coralalgal reef establishment and evolution during the last deglaciation have been well documented through chronological, sedimentological, and paleontological studies, and provide unique data sets upon which past sea-level records have been reconstructed[1–15]. Most of these records display, since the Last Glacial Maximum (LGM), several major intervals of rapid sea-level rise over timescales of several centuries, referred to as melt water pulses, in the uppermost Pleistocene.

Since the early 1930s[16], several deep banks, with crests lying in about 60 mbsl, were known to occur along the south Texas shelf edge (Fig. 1). The coralgal origin of the banks was first proposed in the mid-1970s[17,18] based on five banks from which rock samples were collected by piston coring, dredging, box coring, and Van Veen grab. The rocks consist mainly of dead corals (*Agaricia sp.*, *Madracis sp.*, *Madracis asperula*, *Madracis brueggemanni*, *Madracis myriaster*, and *Paracyathus pulchellus*) and coralline algal nodules. Only two samples were dated; a coral sample from the top of Dream Bank at 68 m yielded a radiocarbon age of $10,580 \pm 155$ years BP ($11,901.5 \pm 335.5$ calendar years BP), and a coralline algal sample from the base of Southern Bank produced a radiocarbon age of $18,900 \pm 370$ years BP ($22,361 \pm 428$ calendar years BP)[17,18]. In late 1990s, a multi-channel seismic grid on one of the reefs, Southern Bank, indicates the thickness of the bank to be about 40–50 m[19]. It is also concluded that the drowned banks along the south Texas shelf edge were established on paleo highs associated with antecedent siliciclastic topographies such as either beach barrier islands or beach ridges developed during late LGM or earliest deglaciation[19]. In absence of detailed chronologic dates and based upon the current water depth range of these bank tops at about 60 mbsl, the demise of these reefs was proposed to have occurred between ~12,250–11,500 Cal BP. Recent studies show that during the LGM, the south Texas coastal system consisted of a bay bounded by the Rio Grande and Colorado lowstand shelf edge deltas, isolated from the open ocean by a barrier island complex[20] (Fig. 1b). The coralgal reefs likely established themselves on top of this lowstand coastal system, thrived, and grew vertically in less

than ~8000 years by tracking the 40–50 m of sea-level rise during the uppermost Pleistocene[19–21]. Ultimately, the south Texas reefs drowned and, starting at ~9 ka, were subsequently partially buried by the Holocene Texas Mud Blanket[17–21] (TMB).

The observed 40–50 m vertical accretion of the coralgal banks in about 8000 years suggests average rates of sea-level rise of 5–6 m per millennium, as in published sea-level records[1,2,12,13]. This pace could have occurred only with the occurrence of scleractinian coral species, including *Acropora palmata* and *Acropora cervicornis*, which display unusually fast growth rates and create the main coral framework of the Caribbean reefs[22]. Although these species are not currently growing at the latitudes of the northern Gulf of Mexico (GoM), except for a few colonies of *A. palmata* newly established at the Flower Garden Banks (FGB) (Fig. 1a) in the past decade[23], it is assumed that these species formed the coral framework of the south Texas shelf edge drowned banks. This assumption is bolstered by the recent discovery that *A. palmata* and *A. cervicornis* grew in large numbers at the base of the FGB[24] as early as 10,200 cal BP, based on radiocarbon dating. The occurrence of these coral species as early as the earliest part of the Holocene in the northern GoM strengthens the inference that they most likely form the coral framework of the uppermost Pleistocene south Texas shelf drowned banks. Additionally, modern and presumably deglacial near-surface circulation patterns in the GoM show that it is and was responsible for carrying biotic communities into the GoM from the Caribbean[25].

It has been established that carbonate production areas shrink through backstepping so to remain within the euphotic zone when responding to sea-level rise; as such coralgal reefs form distinct sets of terraces[26] as they grow vertically keeping up with sea-level rise. During transgressions, therefore, episodic and rapid sea-level rise events result in set of terraces, preserving the nature of sea-level rise and diagnostic morphological features of reefs struggling to keep up with rising sea-level[26–30]. Ultimately, when the area of carbonate production has shrunk to a minimum through systematic backstepping, reefs are unable to grow

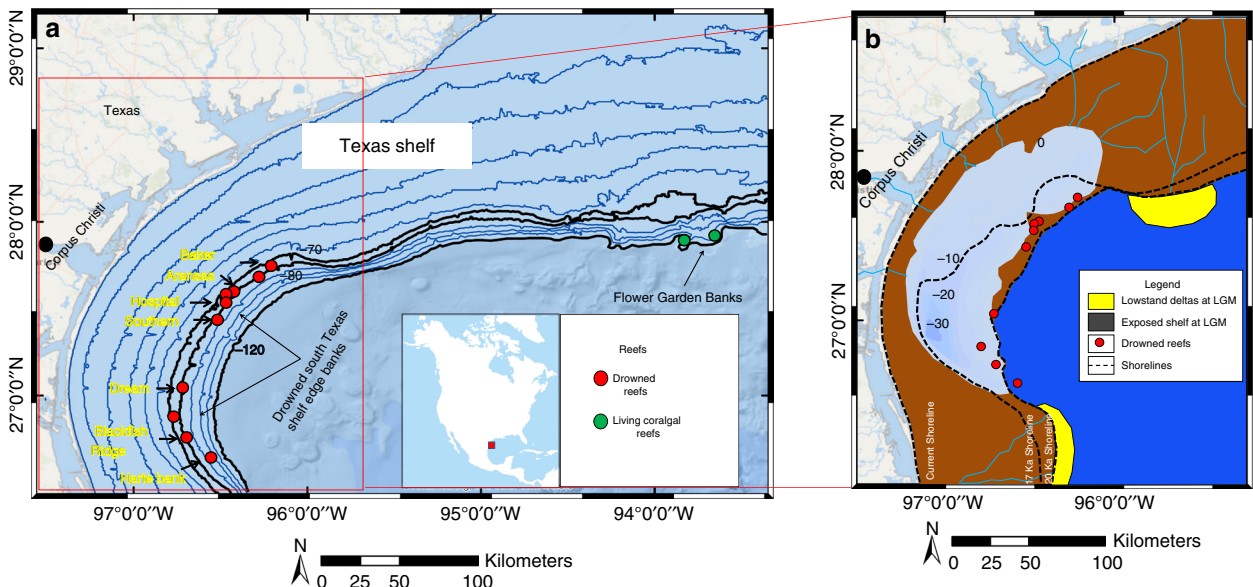

**Fig. 1** Modern and uppermost Pleistocene Texas shelf where living and drowned coralgal reefs are located. **a** Drowned banks along the south Texas shelf edge are shown as red dots. The Flower Garden Banks, living coralgal reefs 150 km south of Galveston Island, are displayed as green dots. Note that, with one exception, the drowned reefs are currently located between 70 and 80 m isobaths. **b** Uppermost Pleistocene south Texas shelf coastal systems at 20 and 17 ka[20] (located in Fig. 1a by red rectangle), illustrate a shallow bay up to 35 m deep, isolated from the open ocean by a barrier island complex, on top of which the south Texas reefs (shown in red dots) were established during the early part of last deglaciation

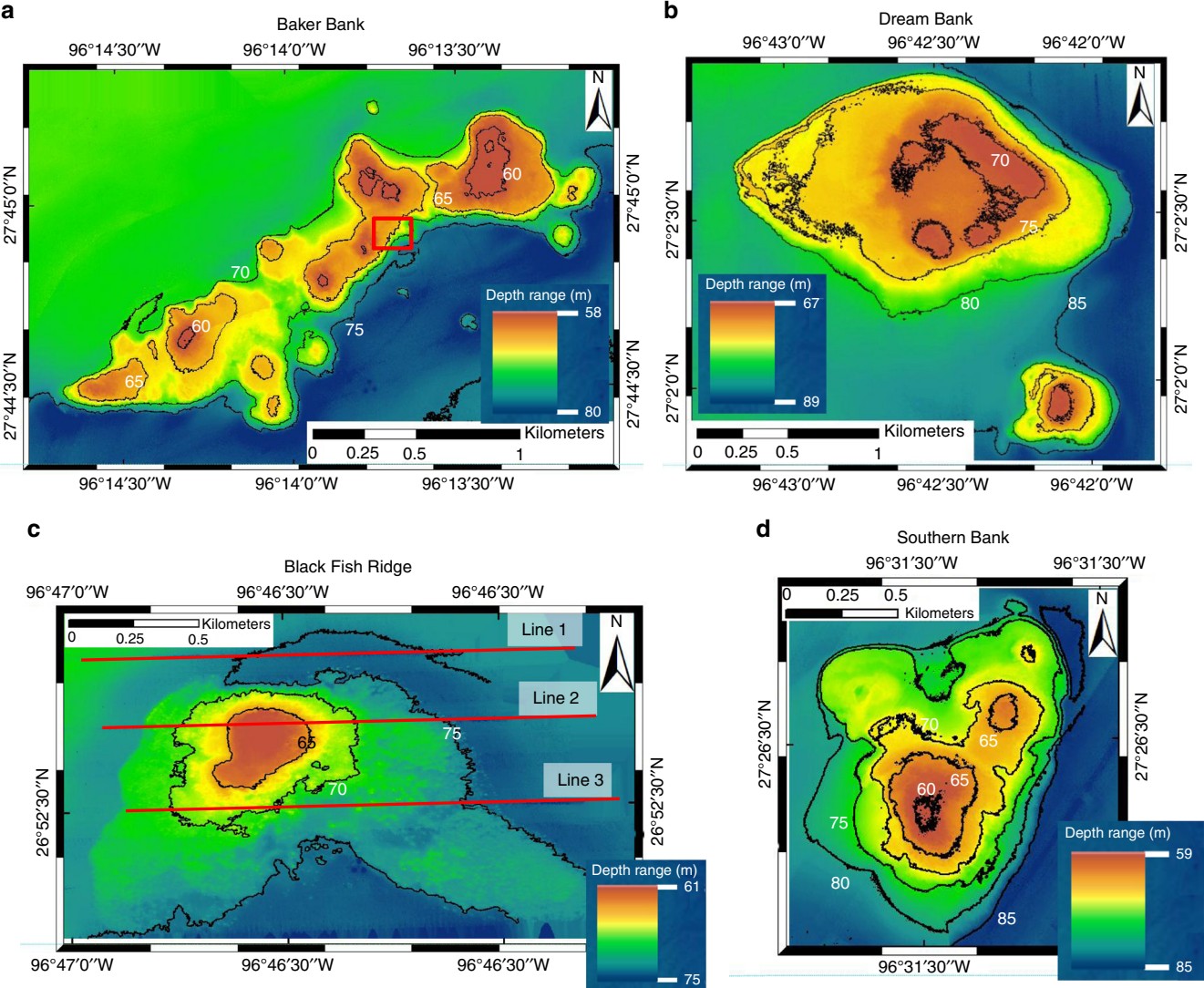

**Fig. 2** High-resolution multibeam bathymetric maps illustrate the morphology of four of the ten imaged drowned coralgal reefs. **a** Baker Bank, **b** Dream Bank, **c** Blackfish Ridge, and **d** Southern Bank (see Fig. 1a for their geographical locations). Spurs (ridges) and grooves (troughs) are usually identified on the south-eastern (windward) side of the banks; the red rectangle in Fig. 2a locates well-developed spurs and grooves shown in Fig. 3a. Dream Bank in Fig. 2b represents distinct atoll morphologies, with well-developed lagoons surrounded by rimmed margins (see Fig. 3c). Dream Bank also clearly displays a set of distinct terraces, illustrated in Fig. 3d, e. Red lines on Fig. 2c locate the three 3.5 kHz seismic lines (1, 2, 3) shown in Fig. 4

vertically fast enough to keep up so as to remain within the euphotic zone and reefs ultimately drown[26,31]. The edifices of drowned reefs sit below the euphotic zone, as the series of drown banks along the south Texas shelf edge, which are no more vertically accreting, although their crests are still covered by live ahermatypic wire corals, sea-fans, mollusks, annelids, bryozoans, and red algae, and are known to be excellent fishing grounds[32].

The new data presented here provides an opportunity to quantify well-imaged back-stepping terraces and identify nature of the sea-level rise during last deglaciation leading to the development of common backstepping morphologies. High-resolution multibeam mapping and seismic profiling of 10 drowned banks, located along a 120-km-long stretch of the south Texas outer shelf, identify six common terrace levels; these identical morphologies provide new opportunities to understand coralgal reef evolution through backstepping and terrace formation, most likely triggered by decade to century-long punctuated sea-level rise during the middle part of last deglaciation. Existing sea-level records do not have the ability to resolve these smaller amplitude variations. Hence, it is pertinent to investigate

geological records that directly document spatiotemporal sea-level changes to determine if decadal to century-scale sea-level rise episodes are common occurrences.

## Results

**True coralgal reef morphologies**. Multibeam bathymetric mapping and 3.5 kHz seismic profiling, acquired in September 2012, onboard the R/V *Falkor* (Fig. 2 and Supplementary Figs. 1–6), showcase the detailed morphological architecture of the south Texas shelf edge drowned banks. Spurs and grooves, typical morphological adaptations to high-energy inner fore reef conditions[33,34] (Figs. 2a and 3a, b), are preferentially observed in the high-resolution bathymetry on the south-eastern margins of several mapped banks and, therefore, coincident with their windward high-energy sides; on their protected north-western lee sides, these features are conspicuously absent. In mid-1970, spurs and groves were already observed, aligned perpendicular to the slope of the bank, by submarine operations using DRV Diaphus[35]. Moreover, the new data presented here provides

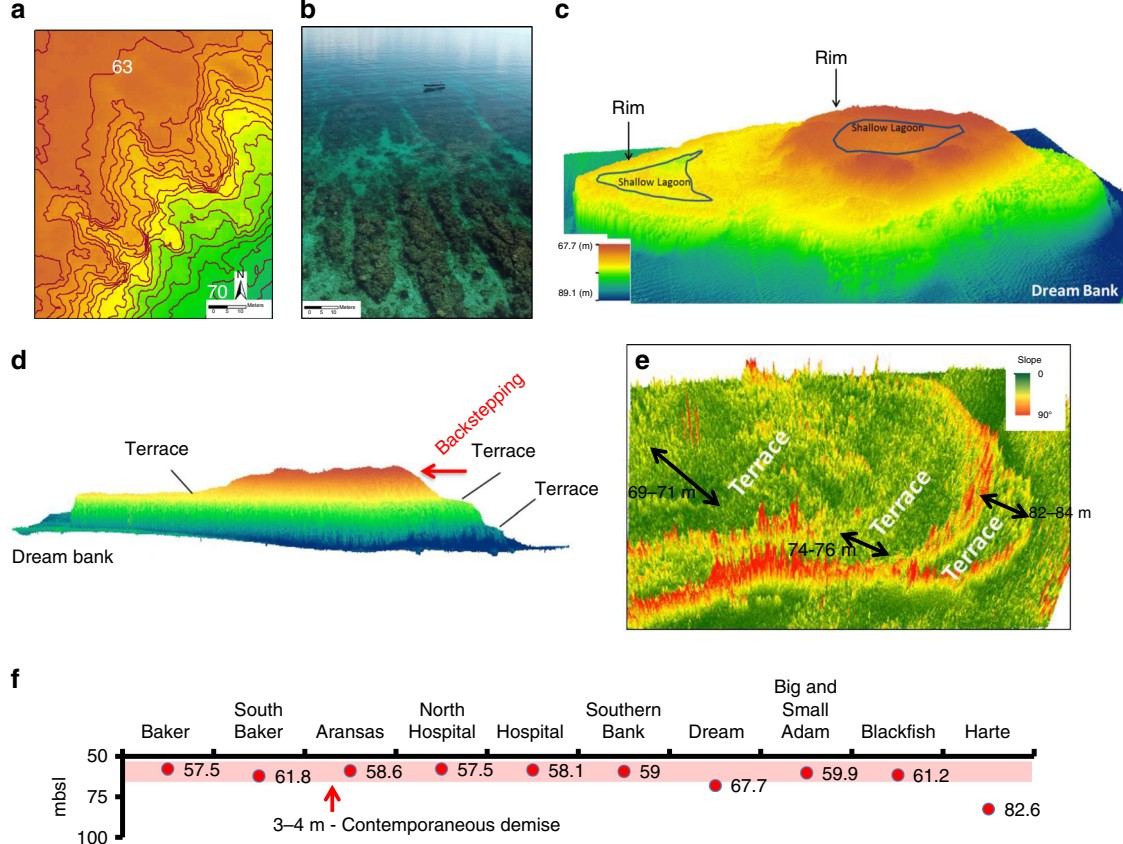

**Fig. 3** True coralgal reef morphologies as evidenced by high-resolution multibeam bathymetric maps. **a** Detailed bathymetry (contour interval 0.5 m) displays a clear example of spurs and grooves on the south-eastern margin of Baker Bank (see Fig. 2a for location), **b** Picture of modern spurs and grooves in front of the Belize Barrier Reef east of Tobacco Range, (Photo by Brandon Martin) as an analog for Fig. 3a; note the similar scales between both fossil and modern spur and grove sets. **c** Oblique bird eye view of Dream Bank displays clear atoll morphologies, rimmed margins enclosing shallow lagoon, at two different levels. **d** Side view of Dream Bank (VE: ×20) displays a series of terraces, characteristic morphology of coralgal reef backstepping in response to punctuated high rates of sea-level rise[26]. **e** Slope angle map for Dream Bank clearly identifies the well-defined terraces and faces (shown in 3D), where red color (steep slopes) represents terrace faces and green color (gentle slopes) terrace flats. **f** Plotted depths to crest of the ten drowned banks, eight of which lie within a 3–4 m-depth range from 57.5 to 61.8 mbsl. Such a narrow depth range testifies to their contemporaneous demise

an opportunity to quantify well-imaged backstepping terraces (Fig. 3c–e), defined as flat areas bounded by steep slopes, common in nine of the ten surveyed banks. These terraces, separated by 1–2 m high faces of coralgal reef rock as was previously observed using submersibles[35], are quantitatively analyzed based on multibeam data. Additionally, Dream Bank displays narrow-rimmed margins enclosing shallow lagoons at two different backstepping terrace levels, typical coralgal atoll morphologies (Fig. 3c).

**Ultimate coralgal reef demise**. The 57.5–61.8 mbsl depth range in which the crests of eight of the ten drowned coralgal reefs occur, point to their contemporaneous demise (Fig. 3f). Furthermore, this depth range coincides with stranded paleo-shorelines and subtidal shoal complexes observed in the GoM[36] (~58 mbsl), Caribbean[37] (~57 mbsl), and Southwest Pacific[37] (~56 mbsl). These paleo-shorelines and shoals are interpreted to have been abandoned by an ~11.5 ka event of rapid rise in global sea level, linked to the onset of MWP-1B occurring at the end of the Younger Dryas[1,15]. It is hypothesized, therefore, that the final demise of the south Texas drowned banks was triggered by the MWP-1B, at ~11.5 ka. The coralgal reefs could not keep up[26,31] with the rapid rise in sea level because their carbonate production surface areas had shrunk to a minimum through systematic

backstepping, as an overall response to the last deglaciation sea-level transgression.

Stressors other than sea-level rise can negatively affect coralgal community growth, such as fluctuations in water turbidity, temperature, and salinity. However, siliciclastic sediment influx into the south Texas shelf edge was minimal during the uppermost Pleistocene transgression[20], when coastlines migrated landward. Initial burial by the TMB was initiated at ~9 ka, and thereby post-date reef drowning by ~2.5 ka. Temperature and salinity likely did not trigger the widespread collapse of the south Texas banks. During the Younger Dryas, sea surface temperatures dropped only by ~1.5 °C to reach 26 °C, and sea surface salinity increased from 34 to 36.5 parts per thousand in the northern GoM[38]. These nominal changes likely did not modify the coralgal reef ecology because during the time period of reef development, sea surface temperatures and salinity are estimated to have fluctuated with an even greater magnitude, between 25 and 29 °C, and 34–38 parts per thousand, respectively[38].

**Terrace hypsometric analysis**. Hypsometric curves, generated from eight banks, identify sets of backstepping terraces at uniform water depths, within a range spanning 75–60 mbsl. Four individual terraces are identified at: $74 \pm 1$, $70.5 \pm 1.5$, $66.5 \pm 1.5$, and $63 \pm 1$ mbsl. The terraces are separated by 2–4-m-high steep face. As imaged in 3.5 kHz seismic lines, a fifth well-developed

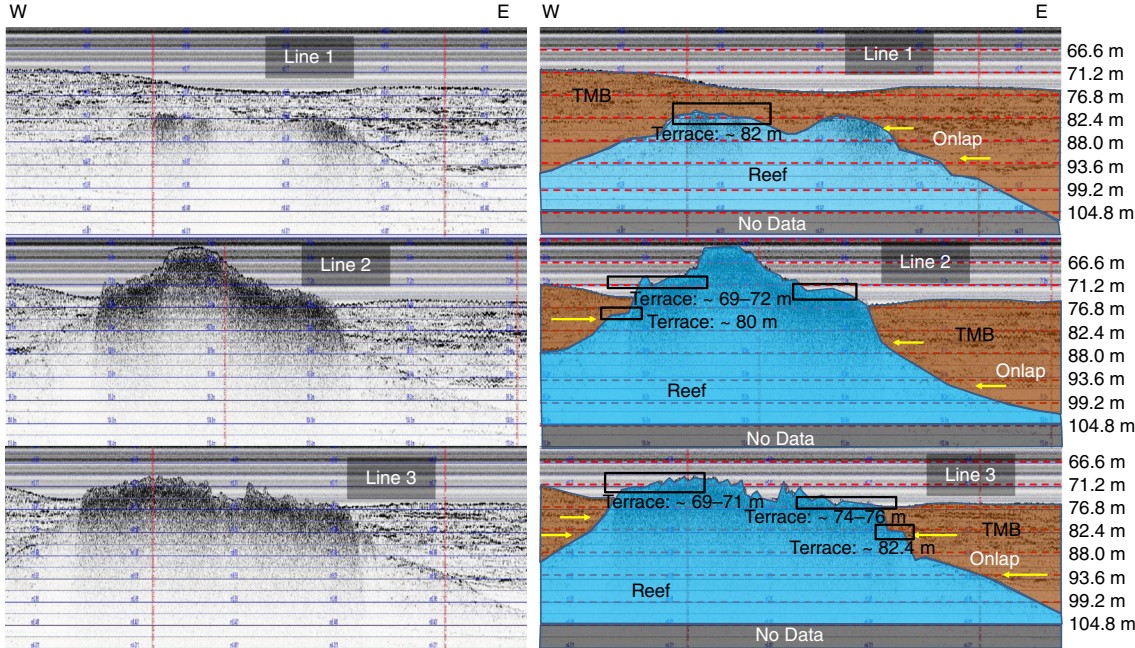

**Fig. 4** A 3.5 kHz Chirp uninterpreted and interpreted lines acquired over Blackfish Ridge Bank. The three seismic lines are located in Fig. 2c. Coralgal reef is colored in blue, the TMB, partially burying the reef, in brown; yellow arrows point to clear onlaps on the reef by the TMB. Three terrace levels, the deepest at 82 mbsl and two shallower terraces at 74–75 and 69–71 mbsl, are identified and are marked by black rectangles

common terrace, buried by the TMB, is identified at $82 \pm 1$ mbsl (Figs. 4 and 5a and Supplementary Figs. 1–6). Moreover, a sixth terrace was mapped at $94 \pm 1.5$ mbsl only on Harte Bank—(the deepest bank with an exposed crest and base at 82 and 102 mbsl, respectively; Supplementary Fig. 7), discovered during the 2012 research expedition aboard the R/V *Falkor* cruise. Because both subsidence and glacio-isostatic adjustment (GIA) rates are assumed to be identical along this 120 km of the south Texas shelf, the observed five common terrace depth ranges can be considered coeval. Despite the absence of systematic chronologic dates for each of the terraces, their consistent depth ranges, among several reefs growing over such this long stretch of the south Texas shelf edge, are interpreted to reflect contemporaneous and systematic backstepping linked to punctuated sea-level rises.

**Paleo terrace depth estimates**. Sea-level changes are dependent on ice-sheet growth and decay, tectonics, and sediment overloading of the shelf and vary in different parts of the world, referred to as relative sea-level (RSL). RSL curves incorporate eustatic sea-level (ESL) fluctuations and it is usually difficult to separate the two (RSL and ESL). The Northwestern GoM is an ideal location for which RSL drivers and their amplitudes are well constrained and provides the opportunity to examine ESL signals. The two main drivers for RSL change in northern GoM are GIA[39] (0.71 mm per year of uplift since 21,000 calendar years BP), and subsidence[40] (0.5 mm per year from past 21,000 years). Considering a linear rate for GIA and subsidence for the last deglaciation, and the current depth of the terraces on drowned banks, the depth of each terrace is recalculated and used as indicator of ESL (Table 1). The corrected depths are compared with an ice-volume sea-level curve[15] to calculate the corrected age range with uncertainties, during which each of the six terraces was developed. These calculations are based on two assumptions: the GIA and subsidence rates are linear, and the development of terraces occurred at sea level. The uncertainties associated with the age model are dependent upon the GIA, subsidence, and relation of

paleo water depth to terrace depth. Calculating GIA and subsidence with corrected age model demonstrates that the deviation in the GIA and subsidence are less than three percent. Further, atoll and spur-groove morphologies, clearly observed in the high-resolution bathymetric data sets, indicate that the reefs, when flourishing, were keeping up with sea level.

**Paleo terrace depths and Greenland climate record**. The corrected depths of the observed six common terrace levels, identified on the south Texas banks, are projected onto a global eustatic sea-level curve[15] (Fig. 5b) and their equivalent ages with uncertainties are estimated based on these projections. Then, these ages with their associated uncertainties are projected onto the NGRIP $\delta^{18}O$ record[41,42]. This climate record from Greenland is, to our knowledge, the only existing high-resolution upper Pleistocene climate record during which the six terraces were formed along the south Texas shelf. The comparison of both records is the only possible opportunity to attempt to understand the cause and effect relationship between warm climate intervals, melting of glaciers (ice-stream/ice-sheet collapse), sea-level rise events, and terrace development. As observed in Fig. 5b, out of the six terraces, four terraces correspond to warm interstadials, one to a stadial–interstadial transition, and one only to a cold stadial. The NGRIP $\delta^{18}O$ record represents climate variations in Greenland. Figure 5b further illustrates that the number of occurrence of the terrace depth zones are similar to the number of warm events observed on NGRIP $\delta^{18}O$ record. Moreover, the correlation of each terrace to a warm interstadial period, with one exception, is noteworthy. These warm periods, therefore, can further be linked to ice-sheet/ice-stream collapse events, causing rapid sea-level rise events of the orders of few meters per century, which lead to the development of common terrace morphologies on south Texas shelf banks.

**Discussion**

In absence of correct chronologic dates, the formation of these terraces, common to nine coralgal reefs, located along a distance

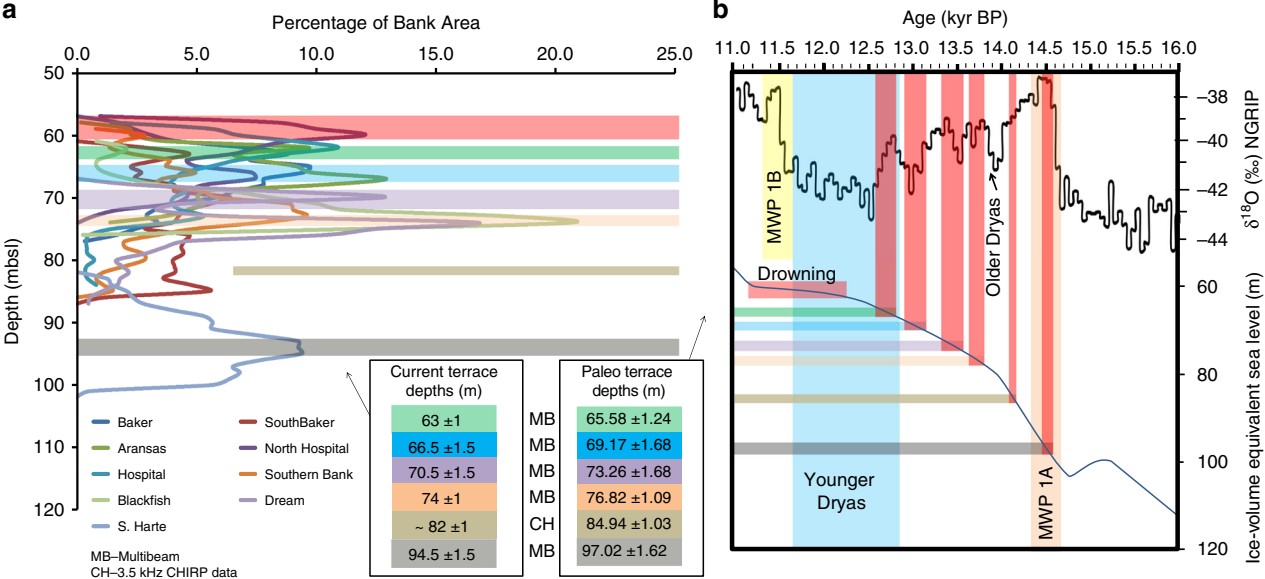

**Fig. 5** Punctuated sea-level rise events over timescales of decades to century based on coralgal reef terrace levels and their connection with warming intervals in the North Greenland Ice Core Project climate record during last deglaciation. **a** Hypsometric curves for nine south Texas shelf drowned banks, based on high-resolution multibeam bathymetry data and 3.5 kHz seismic lines, identify the occurrence of a series of terraces common to the banks. Each depth range (mbsl) of the four common shallower terraces is based on the multibeam data sets, green (63 ± 1 m), blue (66.5 ± 1.5 m), purple (70.5 ± 1.5 m), and orange (74 ± 1 m). Two of those terraces (purple and orange), in addition to a deeper one at 82 ± 1 m, are also identified on the 3.5 kHz seismic lines (Fig. 4). An additional terrace level is identified on the multibeam map of Harte bank (deepest bank): gray (94.5 ± 1.5 m). **b** Paleo terrace depths (Table 1) with uncertainties are projected onto the ice-volume equivalent sea-level curve[15]. The age equivalent of each terrace is projected onto the NGRIP δ18O record[41,42] with associated uncertainties (red vertical bands). The Younger Dryas interval is represented by a blue band. Melt water pulses (MWP) 1A and 1B are represented by vertical light red and yellow bands

**Table 1 Estimated Paleo terrace depths and their age range (see details in the text and methodologies)**

| Depth of terrace | Inferred age from terrace depth (calendar years BP) | GIA (m) | Subsidence (m) | Total depth change (m) | Paleo terrace depth (m) | Inferred age range from Paleo terrace depth (calendar years BP) |
|---|---|---|---|---|---|---|
| 59.25 ± 1.75 (Drowning) | 11,375 ± 275 | 8.07 ± 0.20 | −5.68 ± 0.13 | 2.38 ± 0.33 | 61.63 ± 2.08 | 11,200–12,400 |
| 63 ± 1 | 12,300 ± 200 | 8.73 ± 0.14 | −6.15 ± 0.1 | 2.58 ± 0.24 | 65.58 ± 1.24 | 12,550–12,800 |
| 66.5 ± 1.5 | 12,750 ± 150 | 9.05 ± 0.11 | −6.37 ± 0.07 | 2.67 ± 0.18 | 69.17 ± 1.68 | 12,900–13,150 |
| 70.5 ± 1.5 | 13,150 ± 150 | 9.33 ± 0.11 | −6.57 ± 0.07 | 2.76 ± 0.18 | 73.26 ± 1.68 | 13,300–13,550 |
| 74 ± 1 | 13,475 ± 75 | 9.56 ± 0.05 | −6.73 ± 0.03 | 2.82 ± 0.09 | 76.82 ± 1.09 | 13,650–13,800 |
| 82 ± 1 | 14,025 ± 25 | 9.94 ± 0.02 | −7 ± 0.01 | 2.94 ± 0.03 | 84.94 ± 1.03 | 14,050–14,150 |
| 94 ± 1.5 | 14,400 ± 100 | 10.22 ± 0.07 | −7.2 ± 0.05 | 3.02 ± 0.12 | 97.02 ± 1.62 | 14,450–14,550 |

of over 120 km on the south Texas shelf edge, indicates that during the recent peak deglaciation sea level did not always rise gradually, but rather was characterized by a series of punctuated and rapid sea-level rise events over decades to one century, previously only recognized during late Holocene[43,44]. Because climate warming and resulting ice-sheet collapses have been predicted for the future decades and centuries[45,46], the steady and gradual sea-level rise, observed over the past two centuries may, therefore, not be a complete characterization of how sea level would rise in the future. Furthermore, there is a scientific need to utilize advanced technologies, including high-resolution bathymetry systems combined with systematic drilling of reefs and accurate dating techniques; this study serves as a guide to future research endeavors that seek to inform sea-level rise rate and amplitude projections. Researchers that model sensitivity of sea-level fluctuations—past and present—require as much information as possible regarding smaller amplitude events, and the best place to find this information is from the geological record. The documentation of decades to century-scale

punctuated sea-level rise events with magnitudes of a few meters implies that deglaciation and associated sea-level rise is a non-steady process. Rate of sea-level rise has been observed to accelerate since the past two decades[47]; therefore, these results have significant implications for the community of science researchers that examine sea-level rise past and present, and for how society prepares for coastal flooding and inundation hazards in the coming decades to centuries.

## Methods

**Radiocarbon date calibration.** Calib Rev 7.0.4 was used to calibrate the radio-carbon ages collected in 1970s[17,19,21]. Calibration data set marina13.14c is used with Delta R = −30 ± 9. The new calibrated calendar year ages are 11,901.5 ± 335.5 calendar years BP for the top of Dream Bank and 22,361 ± 428 calendar years BP for the base of Southern Bank. These ages are not incorporated into the age model but are used to only indicate that these reefs grew during last deglaciation.

**Data collection R/V Falkor.** During a 15-day long research cruise in September 2012 onboard R/V Falkor, funded by the Schmidt Ocean Institute, high-resolution multibeam sonar and 3.5 kHz seismic data sets were acquired over 10 drowned

coralgal reefs on the south Texas shelf edge. The research vessel was equipped with state-of-the-art instrumentation, including a Kongsberg EM 710 multibeam echo sounder to collect high-resolution (<0.5 m) bathymetric maps and a high-resolution seabed mapping (3.5 kHz seismic) system, Knudsen CHIRP 3260, to image the sub sea-floor sedimentary units. The ancillary components of the multibeam system included: SeaPath 320 heading, attitude, and positioning sensor, CNAV positioning correction service, and Valeport SV profiler. Multibeam data was processed using Caris 7.1 and further imported to Arc G.I.S. 10.1 to build and investigate the bathymetric maps of these drowned reefs. The CHIRP data was analyzed utilizing the Echo Post Survey software.

**Hypsometric curves—data analysis**. Hypsometric curves are generated for nine of the ten drowned coralgal reefs (Fig. 5a). Bathymetry data sets are clipped into subdata sets encompassing each individual drowned reef. The total surface area of each reef is divided into 1 m-depth intervals and the surface area of each interval is calculated by the number of pixels (each one representing one square meter). To create a hypsometric curve for each individual reef, the percentage of each one meter depth interval is determined using their calculated surface area. Each peak in a given reef's hypsometric curve represents individual terrace. When the nine hypsometric curves are plotted together, overlapping peaks identify common terrace depth zones. For each common terrace depth zone, a median terrace depth is determined. Depth uncertainties are evaluated as the difference between the median terrace depth values and their maximum or minimum depth range.

**Computing paleo terrace depth**. In the absence of chronologic dates, the current depths of the terrace zones, ice-volume sea-level curve[15], in addition to GIA[39] and subsidence[40] rates, are used to estimate paleo terrace depths. First, the terrace depth zones (with depth uncertainties) are compared with an ice-volume sea-level curve[15] to identify the age range (including uncertainties) for the development of each terrace zone. To estimate the paleo terrace depth, depth change for each terrace due to GIA and subsidence is calculated by multiplying the estimated age range of each terrace depth zone with avg. rate of GIA[39] (0.71 mm per year) and subsidence[40] (0.5 mm per year). The total change in depth is calculated by adding GIA and subsidence (uplift is considered positive and subsidence is considered negative). The estimated total depth change is added to the current terrace depth to identify the paleo terrace depth for each terrace. The paleo terrace depths are further compared with an ice-volume sea-level curve[15] to estimate the age range for the development of each paleo terrace.

Two assumptions are included in the analyses: the rates of subsidence[40] (0.5 mm per year) and GIA[39] (0.71 mm per year) are considered linear and 95% probability ice-volume sea-level curve[15] is used.

**Data source**. Digital data generated by the environmental sensor systems onboard R/V *Falkor*, including multibeam, are archived and freely available to the public via Rolling Deck to Repository and given Digital Object Identifiers—http://www.rvdata.us/catalog/FK005B

**Data availability**. All relevant data are available from authors.

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

## Acknowledgements

The data collected in this study are based upon work supported by the Schmidt Ocean Institute during cruise FK005B aboard R/V *Falkor*. We thank the Schmidt Ocean Institute and the R/V *Falkor* officers, crew members, and onboard technicians. We are grateful to Jean Aroom at the Rice University G.I.S. Datacenter for training the senior author with advanced Arc G.I.S. applications..

## Author contributions

P.K., A.W.D., J.W.T. and T.C.S. participated on the R/V *Falkor*; P.K. and J.A.N. processed the data; P.K., A.W.D. and J.A.N. co-wrote the manuscript; P.K., A.W.D., J.A.N., J.W.T. and T.C.S. edited the manuscript.

## Additional information

**Competing interests:** The authors declare no competing financial interests.

