## [Peer Review File · Nature Communications]

Reviewers' comments:

Reviewer #2 (Remarks to the Author):

The message of the manuscript is that the last deglaciation was not a gradual between ~14 – 12 kyrs, which is between the melt-water pulse (MWP) 1A and 1B, but occurred in pulses, each producing a rapid sea level rise. This conclusion is drawn from recently acquired multibeam bathymetry and CHIRP sonic data that image the drowned reefs and their terraces. The backstepping of the reef margin formed a terrace, which is convincingly interpreted as the result of pulses of sea level rise. The age dates for the reef growth and demise are from dredge samples published in 1976. The manuscript does not specify on which of the drowned reefs these dredges were made. As a consequence the exact timing of the terrace formation cannot be established but is inferred from the depth of the terrace and the comparison to the sea level curve from Tahiti. Despite these limitations in the data the paper presents evidence of a pulsed deglaciation in the early stages of the last sea level rise.

Unfortunately the author(s) have embroidered or muddled this clean story with issues that do not contribute or even raise doubts about the main findings. For example, the long justification that the imaged features are indeed reefs is troublesome. Either the dredge samples on which the age dates are based on are coralgall material or they are not. The authors do not say in the manuscript. Instead evidence of coral growth is taken from the Flower Gardens reefs that are 200 km further east.

It also needs to be mentioned that Droxler and his students have published several papers about these drowned reefs and always called them coral reefs and placed them into the same time frame as this study. The only new data here is the high-resolution bathymetry that images the terraces, indicating the pulsed sea level rise.

This is an important detail and has consequences for the understanding of past and future sea level fluctuations. This new finding could be documented in a much more concise fashion and I would encourage the authors to focus the manuscript, refer to the papers of Belopolski and Droxler (1999) and Droxler and Jorry (2013) for the overall relationship between reef growth and deglaciation.

Some minor points are the awkward title of the manuscript that does not convey the main message of the manuscript. Likewise, I had to read the abstract twice before I recognized the main as it focuses first on the description/interpretation of the reef morphology before talking about pulsed sea level rise. Instead of saying that reefs are dipsticks in the ocean, the opening sentence should say that reefs can monitor rates of sea level rise and that here a study is presented that indicates that during deglaciation pulses of sea level rise are more likely than a steady sea level rise.

Reviewer #3 (Remarks to the Author):

This paper presents new high-resolution mapping of drowned reefs along the shelf to the south of Texas. Given the depth of these coral terraces relative to modern sea level, the features possibly represent interrupted jumps in sea-level rise related to deglacial meltwater pulses. If true, these results provide an important new record of deglacial sea-level during the latest deglaciation.

However, for these punctuated sea-level jumps to be relevant in our understanding of deglacial sea-level rise, these terraces must be associated with a robust chronology to demonstrate the relationship with known meltwater pulses (MWP). For this paper, the lack of independent chronological constraints on the coral terraces is problematic. The authors link terrace depth with the well-dated sea-level record from Tahiti (Deschamps et al., 2012). The problem here is that both sea level records (Texas and Tahiti) are measurements of RELATIVE sea level change, not EUSTATIC (global mean) sea level change. Certainly there is a eustatic signal in both records, but

RELATIVE sea level at any given site also reflects changes in a number of local effects such as tectonic uplift/subsidence, glacial isostatic adjustment (GIA; Peltier, 1999; Love et al., 2016), meltwater source and "sea-level fingerprinting" (Clark et al., 2002), etc. As such, relative sea level may diverge substantially from eustatic sea-level (Peltier, 1999; Lambeck and Chappell, 2001; many other examples of GIA adjustment). Since relative sea level is spatially varying, connecting two independent sea-level records by their local sea level change is likely not appropriate because it would assume that local effects are negligible.

Therefore, the chronology used in this manuscript may not be correct because of the relative/eustatic sea level differences. The conclusions of this paper related to the timing of sea level jumps may not be defensible because they rely on this incorrect linkage of relative sea level records.

If the authors can appropriately correct their modern depths for tectonic adjustment, GIA, and MWP fingerprinting (all drivers of local sea-level change), then they may be able to connect their depths with the chronology of global mean sea level (e.g Lambeck et al., 2014). However, such a correction will require a large number of assumptions that may be indefensible.

In light of this major limitation in chronology, I cannot recommend this manuscript for publication in its current form.

References

Clark, P.U., et al., 2002, Sea-level fingerprinting as a direct test for the source of global meltwater pulse 1A: *Science*, v. 295, 2438-2441.

Deschamps, P., et al., 2012, Ice-sheet collapse and sea-level rise at the Bolling warming 14,600 years ago: *Nature*, v. 483, 559-564.

Lambeck, K. and Chappell, J., 2001, Sea level change through the last glacial cycle: *Science*, v. 292, 679-686.

Lambeck, K., et al., 2014, Sea level and global ice volumes from the Last Glacial Maximum to the Holocene: *Proceedings of the National Academy of Sciences*, v. 111, 15296-15303.

Love, R., et al., 2016, The contribution of glacial isostatic adjustment to projections of sea-level change along the Atlantic and Gulf coasts of North America: *Earth's Future*, v.4, 440-464.

Peltier, W.R., 1999, Global sea level rise and glacial isostatic adjustment: *Global and Planetary Change*, v. 20, 93-123.

Reviewer # 2

Comment 1: The message of the manuscript is that the last deglaciation was not a gradual between ~14 – 12 kyrs, which is between the melt-water pulse (MWP) 1A and 1B, but occurred in pulses, each producing a rapid sea level rise. This conclusion is drawn from recently acquired multibeam bathymetry and CHIRP sonic data that image the drowned reefs and their terraces. The backstepping of the reef margin formed a terrace, which is convincingly interpreted as the result of pulses of sea level rise.

Response: Indeed, this is the primary main theme of the manuscript; namely, the acquisition of high-resolution bathymetric images, when combined with CHIRP sonar data, provide convincing evidence that these drowned reefs along the western margin of the Gulf of Mexico maintain terraces at consistent elevations, so as to indicate coeval backstepping. Our transformation of space for time, using the global sea level curve, indicates that this could arise due to punctuated sea-level rise events occurring during last deglaciation. We are happy that the primary hypothesis is well received by the Reviewer #2.

Comment 2: The age dates for the reef growth and demise are from dredge samples published in 1976. The manuscript does not specify on which of the drowned reefs these dredges were made. As a consequence the exact timing of the terrace formation cannot be established but is inferred from the depth of the terrace and the comparison to the sea level curve from Tahiti. Despite these limitations in the data the paper presents evidence of a pulsed deglaciation in the early stages of the last sea level rise.

Response: The age dates, and sample information (including depth and location of recovery) is now added to the revised manuscript. We note here that only two dates are available: the first is a coral sample from Dream Bank, collected at a depth of 68 m below sea level, with a radiocarbon date $10,580 \pm 155$ years BP (12,280 calendar years BP). A second sample of coralline algae was collected from the base of Southern bank; the radiocarbon analysis yielded an age of $18,900 \pm 370$ years BP (21,480 calendar years BP).

The exact timing of the terraces is inferred in this modified manuscript by correcting the terrace depths for relative sea level changes and comparing with the Ice-volume sea-level curve from Lambeck et al. (2014). With these new changes the approximate timing of the terrace development has been established and comparison to the Ice-volume equivalent sea-level from Lambeck et al. (2014) indicates that the warming events recorded in the NGRIP ice core $\delta^{18}\text{O}$ record have been preserved as punctuated sea-level rise events on the south Texas shelf banks.

Comment 3: Unfortunately the author(s) have embroidered or muddled this clean story with issues that do not contribute or even raise doubts about the main findings. For example, the long justification that the imaged features are indeed reefs is troublesome. Either the dredge samples on which the age dates are based on are corallgal material or they are not. The authors do not say in the manuscript. Instead evidence of coral growth is taken from the Flower Gardens reefs that are 200 km further east.

Response: The first studies of this system were conducted in 1970's, whereby a submersible was used to observe coral rubble distributed on and around the bank (Rezak and Bright, 1976;

Lindquist, 1978). This work also noted channels in the reefs up to a meter deep and a few meters wide, which are the spurs, grooves, and terraces that we identify with the high-resolution bathymetric data presented in our manuscript. Belopolsky and Droxler (1999), and Droxler and Jorry (2013) used only multi-channel seismic to address that the reefs are coralgal.

Reason behind introducing Flower Garden Banks is to provide evidence, as how the south Texas shelf reefs grew upto 40-50 m in less than 8000 years. The growth rates are exceptionally high but presence of *Acropora Palmata* or *Acropora Cervicornis* (the main reef framework builders in the Caribbean currently) has never been identified from any dredged samples collected from the south Texas shelf drowned banks. The main question now is whether these coral species were ever present in the northern part of Gulf of Mexico or not. To answer this, we focused on a living coral colony, Flower Garden Banks, east of the drowned South Texas shelf banks, where *Acropora* species were identified from the base of the reefs, dated to be around ~10,000 years BP indicating that the *Acropora* were present in the northern part of Gulf of Mexico in the past and could rather well be the framework builders for the drowned banks on south Texas shelf under study. Hence it is pertinent to discuss about the Flower Garden Banks.

Comment 4: It also needs to be mentioned that Droxler and his students have published several papers about these drowned reefs and always called them coral reefs and placed them into the same time frame as this study. The only new data here is the high-resolution bathymetry that images the terraces, indicating the pulsed sea level rise. This is an important detail and has consequences for the understanding of past and future sea level fluctuations.

Response: Belopolsky and Droxler (1999) and Droxler and Jorry (2013) defined these drowned reefs as corallgal on the basis of seismic data. We leverage these interpretations to produce the succinct and critically important new messages in our current manuscript: that the morphologies of these reefs, resolvable only with the advent of multibeam technology, indicate conditions of pulsed sea level rise during the recent deglaciation. This interpretation has consequences for evaluating the past and future sea level changes (please see our response to comment 1 above for further elaboration of the contributions of this manuscript).

Comment 5: This new finding could be documented in a much more concise fashion and I would encourage the authors to focus the manuscript, refer to the papers of Belopolsky and Droxler (1999) and Droxler and Jorry (2013) for the overall relationship between reef growth and deglaciation.

Response: As advised by the Reviewer, we have modified the manuscript and now indicate how the previous findings of Belopolsky and Droxler (1999) and Droxler and Jorry (2013) support our current study.

Comment 6: Some minor points are the awkward title of the manuscript that does not convey the main message of the manuscript. Likewise, I had to read the abstract twice before I recognized the main as it focuses first on the description/interpretation of the reef morphology before talking about pulsed sea level rise. Instead of saying that reefs are dipsticks in the ocean, the opening sentence should say that reefs can monitor rates of sea level rise and that here a study is presented that indicates that during deglaciation pulses of sea level rise are more likely than a steady sea level rise

Response: We thank the reviewer for this feedback. The title, abstract, and the manuscript have been revised so to clearly and concisely convey the findings of our research.

Reviewer # 3

Comment 1: This paper presents new high-resolution mapping of drowned reefs along the shelf to the south of Texas. Given the depth of these coral terraces relative to modern sea level, the features possibly represent interrupted jumps in sea-level rise related to deglacial meltwater pulses. If true, these results provide an important new record of deglacial sea-level during the latest deglaciation.

Response: Our manuscript addresses this bold new idea, and the implications of episodic rather than gradual sea-level rise are vast. As described in our manuscript, the fact that 10 drowned banks all possess common terraces at similar depths links their origins to episodic sea-level rise events. We sought to evaluate all other causes that could produce uniform reef demise and terrace formation (e.g., salinity and temperature changes, sediment input) and after ruling these factors out, we are left with the basic explanation that punctuated events of sea-level rise produced reef backstepping and therefore the observed terraces (i.e., this interpretation defaults on the principle of Occam's razor). The translation of space for time by comparing the elevation of these terraces to the global sea-level curve indicates that the possible timing for the formation of these terraces coincides with the record of $\delta^{18}\text{O}$ during warm intervals, and links the episodic sea level rise events and melting of polar ice with the terrace formation. These insights should be

of value to the scientific communities who seek to evaluate past and future conditions of sea-level response to warming.

Comment 2: However, for these punctuated sea-level jumps to be relevant in our understanding of deglacial sea-level rise, these terraces must be associated with a robust chronology to demonstrate the relationship with known meltwater pulses (MWP). For this paper, the lack of independent chronological constraints on the coral terraces is problematic.

Response: We agree that robust chronology could help demonstrate a cause-and-effect relationship. However, these data are not available. Instead, our study aims to support the hypothesis that the last deglaciation was characterized by punctuated, rapid sea level rise events, based on several lines of evidence. Firstly, the consistent occurrence of common terrace elevations and coral morphologies distributed over ~ 100 km of the western Gulf of Mexico. Secondly, after correcting for GIA and subsidence, and trading space for time via the global sea-level record, the fact that the elevations of the common terraces synchronize with excursions of $\delta^{18}\text{O}$ trends. At minimum, these correlations provide substantial evidence to bolster our hypothesis. As far as we are aware, this is the first such evidence provided from the geological record to support the notion that sea-level rise could be event-based, on the vertical scale of decimeters to meters, rather than gradual, as is typically assumed based on the measured record of the past several decades. Drilling these terraces so as to produce samples for robust chronological control is indeed ideal, but the data to date do not exist. We must therefore rely upon alternative means to address the occurrence of common elevation terraces.

Comment 3: The authors link terrace depth with the well-dated sea-level record from Tahiti (Deschamps et al., 2012). The problem here is that both sea level records (Texas and Tahiti) are measurements of RELATIVE sea level change, not EUSTATIC (global mean) sea level change. Certainly there is a eustatic signal in both records, but RELATIVE sea level at any given site also reflects changes in a number of local effects such as tectonic uplift/subsidence, glacial isostatic adjustment (GIA; Peltier, 1999; Love et al., 2016), meltwater source and “sea-level fingerprinting” (Clark et al., 2002), etc. As such, relative sea level may diverge substantially from eustatic sea-level (Peltier, 1999; Lambeck and Chappell, 2001; many other examples of GIA adjustment). Since relative sea level is spatially varying, connecting two independent sea-level records by their local sea level change is likely not appropriate because it would assume that local effects are negligible. Therefore, the chronology used in this manuscript may not be correct because of the relative/eustatic sea level differences. The conclusions of this paper related to the timing of sea level jumps may not be defensible because they rely on this incorrect linkage of relative sea level records. If the authors can appropriately correct their modern depths for tectonic adjustment, GIA, and MWP fingerprinting (all drivers of local sea-level change), then they may be able to connect their depths with the chronology of global mean sea level (e.g. Lambeck et al., 2014). However, such a correction will require a large number of assumptions that may be indefensible. In light of this major limitation in chronology, I cannot recommend this manuscript for publication in its current form.

Response: Reviewer 3 is correct that Texas and Tahiti sea level are measurements of the relative sea level change and not eustatic sea level change. To accommodate the Reviewers comment, we have modified our analysis. The current depths of the terraces are corrected for Relative Sea-

Level changes based on GIA (Milne and Mitrovica, 2008) and subsidence (Winker, 1979).

Comparing the depth of terraces with the Ice volume sea-level curve (Lambeck et al., 2014) – age of each terrace is identified (This is an assumption to identify the approximate total uplift or subsidence in the absence of absolute chronologic dates). The age of each terrace is multiplied with the GIA and subsidence (assumption that the rate was linear in absence of absolute chronologic dates) to identify the total depth/uplift since the development of that terrace. (The GIA is causing an uplift which is considered positive and the subsidence is considered negative). The total change in depth is added to the current depth of each terrace to correct for RSL changes and to identify eustatic sea-level changes. The corrected depths of the terraces are then projected on the ice volume sea level curve of Lambeck et al. (2014).

Reviewers' comments:

Reviewer #2 (Remarks to the Author):

This revised manuscript was improved in regards to several points. It is clearer, more concise and delivers the message early on. In regards to data, the authors now provide evidence of the coralgal nature of the observed banks, and give the age dates of samples collected in the seventies on these drowned reefs. In addition, the authors make an effort to take the global isostatic adjustment (GIA) into account to relate their relative sea level depth given by the rock record to the eustatic sea level change. By doing so the authors are also refining the proposed ages of the punctuated sea level changes.

All this improvement does, however, not completely eliminate the lack of precise chronology of the sea level events. The ages from two locations of these reefs are not incorporated into the age model but serve merely as proof that these reefs grew indeed during the last deglaciation. The older age of 18900 yrs BP from the base of the Southern Bank seems to contradict their age model, which starts at 14.5 kyrs; the younger, however, is close to the proposed drowning age of the "Dream" Bank and thus supports their age assignment of the top drowning. As the authors say, the data for a precise chronology is not there and would have to be obtained by drilling these edifices as the authors forcefully point out at the end of the paper. As such the paper is a perfectly written working hypothesis for a drilling campaign that has very compelling evidence that a test would be positive. Considering the fact that drilling these reefs are years or maybe decades in the future I recommend publication. The main reason is the the authors have a sedimentary record that shows a punctuated sea level rise during the early stages of the last deglaciation. The paper would need the age dates to be make the cause and effect link robust. However, the process of punctuated sea level rise is solid. In addition, the scientific community should be made aware that sedimentary systems, like carbonate build-ups contain a formidable archive of sea level change that strengthens the proxy commonly used for reconstructing past sea level changes.

Some minor points.

In the letter the authors claim that this is "the first record" of the punctuated sea level rise during the last glaciation. This seems not exactly accurate. Lidz and Shinn, 1991, already had mapped drowned reefs, admittedly of younger ages in the Holocene, on the Florida ramp. (Barbara H. Lidz and Eugene A. Shinn; Paleoshorelines, Reefs, and a Rising Sea: South Florida, U.S.A. Journal of Coastal Research Vol. 7, No. 1 (Winter, 1991), pp. 203-229.) The same was reported from clastic deltas by Törnqvist and Hijma in Nature Geosciences 2012, who give various other examples. Maybe some of these works could be cited here.

The title is still awkward; why is there colon and why do you need events? I would simply say "Coralgal Reef Morphology Records Evidence of Punctuated Sea-Level Rise during Last Deglaciation."

The figures still could be improved:

First, all the A B etc labels are too big and have too large of a white background that distract from the figure itself.

Second, all the bathymetry coloring should be in the same range. Currently all banks in Figure 2 have a different depth range.

Third, Schlager's backstepping model could be eliminated from Figure 3 and the backstepping arrow could be placed on Fig. 3 D

Reviewer #3 (Remarks to the Author):

Review of “Evidence of Punctuated Sea-Level Rise Events during Last Mid-Deglaciation”

This manuscript presents new high-resolution bathymetric and seismic data of drowned reefs in the Gulf of Mexico south of Texas. These new maps indicate the existence of extensive terraces at modern depths between 102 and 60 meters below sea level (mbsl). These terraces suggest stabilization of sea level at intervals since the last glacial maximum, likely punctuated by periods of more rapid sea level rise that led to the cessation of coralgal reef growth. Such rapid jumps in sea level are consistent with our understanding of the nonlinearities in the deglaciation of continental ice sheets and associated meltwater pulses (e.g. MWP-1a, MWP-1b, etc), which have been demonstrated in other sea level records.

The high-resolution of these mapped terraces is a novel accomplishment, indicating that the connection between deglaciation and sea level rise may include finer scale MWP events that have not yet been resolved until now. However, the lack of direct chronological constraints on these terraces makes the connection with deglacial events challenging. This is a revised manuscript, and the authors have taken steps to improve their estimates of paleo sea-level by converting the modern depths to a eustatic component, following with a comparison with the Lambeck et al. (2014) record to estimate the timing of sea level jumps. However, I think there are a number of points that need to clarification in the text, to determine what the authors have projected their data onto the Lambeck et al. curve and the NGRIP d18O record. In particular, I think the manuscript underestimates uncertainties in these projections, and I would like to see a more complete inclusion of these uncertainties. In the end, such uncertainties may preclude the connection of these terraces with particular climate events, limiting some of the conclusions of this paper.

Specific comments

Line 38 (and throughout) – Calibrated radiocarbon dates (calendar years) should include uncertainty. In addition, please indicate the calibration dataset you are using.

Line 41 – “This study further point...” should be “This study further points...”

Line 86-88 – Is there a particular range of depths in which the coralgal reefs can survive (photic ‘zone’)? The caption of Figure 3 groups 8 terraces that are within 3-4 depth of a particular depth range. Is this coralgal reef habitat depth uncertainty? If so, shouldn’t this be included as a source of uncertainty in your estimates of paleo sea level from these terraces?

Line 95 – You reference Fig. 4A. There is no Fig 4A. Do you mean Fig 3A?

Line 110 – First usage of GIA. Must define.

Line 114 – remove the word “simply”. Also, why is “keep up” in quotes?

Line 115 – comma before “because” is unnecessary

Line 118 – remove words “to be sure”

Line 121 – “Texas mud blanked” should be “Texas mud blanket”. Also, this is the second time you define TMB.

Line 132-133 – I do not understand this sentence. What is the word “become” referring to? Also, what is the “latter”?

Line 135 – define “GoM” if you intend to use it throughout.

Line 136 – Need to include units of “years” after “21,000”

Line 137-139. The process of estimating the GIA/subsidence corrections should be done iteratively. From what I understand in the text, the initial depth is linked with time using Lambeck et al. This time component is necessary to estimate the GIA/subsidence correction. But the eustatic/subsidence correction changes the implied projection onto time using Lambeck et al. Thus, a new estimate of time must be estimated from Lambeck to provide an appropriate estimate of GIA/subsidence. This is a major challenge for this manuscript. Since there is no direct chronology on the terrace, such an iterative projection of GIA/subsidence correction might lead to uncertainties. The authors should discuss their assumptions in this approach.

Also, in this projection of sea level onto time using Lambeck, are you including depth uncertainty in these projections? There should be uncertainty from measurements (most depths in Table 1 have uncertainty of +/- 1 to 1.5 m), corallgal reef habitat depth range, eustatic uncertainty (from Lambeck et al.). These uncertainties must be included in terrace depth with time. Table 1 implies that the estimates of time for these terraces have no uncertainty. This is disingenuous.

Line 139 – “an indicator of RSL” – By including GIA and subsidence in your correction, aren’t you providing an estimate of the eustatic sea level change that the terraces indicate?

Line 142-144 – This sentence is not clear. What is being projected onto what?

- Are you projecting terrace depths onto ice volume? Or are you projecting terrace depths onto NGRIP?
- Also, how does projecting anything onto NGRIP “identify the eustatic signal” as indicated in the first part of the sentence? NGRIP is not a pure eustatic signal, without some correction for $\delta^{18}\text{O}$ changes due to changes in temperature and precip.
- Are you projecting modern terrace depth onto NGRIP or eustatic corrected terrace depth onto NGRIP? Your table 1 makes it seem as though you are inferring ages using the raw terrace depth, not the eustatic corrected depth. This is not right.
- Are you including depth uncertainty in these projections?

Line 144 (and throughout) – the 18 in $\delta^{18}\text{O}$ should be superscript

Line 144-148. This statement suggests that the sea level jumps line up with warm periods (interstadials) in the NGRIP. There are a number of problems with this statement:

- As stated elsewhere in my comments, I think depth and chronological uncertainties are underestimated in this analysis. Therefore, the projection of sea level rise onto the NGRIP curve should be displayed as a band, NOT an individual line. Such uncertainty may cause this projection to span an interstadial and a neighboring stadial, thereby precluding the connection between the terraces and NGRIP warm periods.
- The NGRIP $\delta^{18}\text{O}$ curve is not solely temperature. This curve also includes source effects from precipitation. See LeGrande and Schmidt (2009, Climate of the Past)
- The two youngest projections (one at ~ 12.5 ka, another at ~ 12.9 ka) in Figure 5 appear to link up with stadials (cool periods, not warm periods) in the NGRIP record.

Line 148 – what is meant by “the cause and effect circle”. Be more specific.

Line 153 – what do you mean by “rapid”? Can you estimate the rate of sea level rise? Related to this question, what is the rate of sea level rise necessary to outpace coralgal reef growth.

Line 154 – what is meant by a “non-steady process at a human time scale”?

Line 154-156. Awkward sentence. Revise.

Line 175. Should there be a hyphen in “sub sea floor”?

Figure Caption 2.

- Please define what is meant by “spurs” and “grooves”
- “well developed” should be hyphenated\

Figure Caption 3.

- “south-western margin of Baker Bank”. Do you mean “southeastern”?
- It is strange to have a URL in the middle of a caption. Is this URL the citation for the photo? Also, what happens if the URL becomes inactive over time?

Figure 5A. What do the abbreviations “MB” and “CH” stand for in the lower right inset?

Reviewer # 2

Comment 1: This revised manuscript was improved in regards to several points. It is clearer, more concise and delivers the message early on. In regards to data, the authors now provide evidence of the corallgal nature of the observed banks, and give the age dates of samples collected in the seventies on these drowned reefs. In addition, the authors make an effort to take the global isostatic adjustment (GIA) into account to relate their relative sea level depth given by the rock record to the eustatic sea level change. By doing so the authors are also refining the proposed ages of the punctuated sea level changes.

All this improvement does, however, not completely eliminate the lack of precise chronology of the sea level events. The ages from two locations of these reefs are not incorporated into the age model but serve merely as proof that these reefs grew indeed during the last deglaciation. The older age of 18900 yrs BP from the base of the Southern Bank seems to contradict their age model, which starts at 14.5 kyrs; the younger, however, is close to the proposed drowning age of the “Dream” Bank and thus supports their age assignment of the top drowning. As the authors say, the data for a precise chronology is not there and would have to be obtained by drilling these edifices as the authors forcefully point out at the end of the paper. As such the paper is a perfectly written working hypothesis for a drilling campaign that has very compelling evidence that a test would be positive. Considering the fact that drilling these reefs are years or maybe decades in the future I recommend publication. The main reason is the the authors have a sedimentary record that shows a punctuated sea level rise during the early stages of the last deglaciation. The paper would need the age dates to be make the cause and effect link robust. However, the process of punctuated sea level rise is solid. In addition, the scientific

community should be made aware that sedimentary systems, like carbonate build-ups contain a formidable archive of sea level change that strengthens the proxy commonly used for reconstructing past sea level changes.

Response: As mentioned by Reviewer #2, the primary focus of the paper is to inform the scientific community of regarding the strong evidence of recorded and preserved punctuated sea level rise events during last deglaciation by shallow marine carbonate systems. We are glad that the central message of the manuscript is well received and recommended for publication by Reviewer #2.

Comment 2: In the letter the authors claim that this is “the first record” of the punctuated sea level rise during the last glaciation. This seems not exactly accurate. Lidz and Shinn, 1991, already had mapped drowned reefs, admittedly of younger ages in the Holocene, on the Florida ramp. (Barbara H. Lidz and Eugene A. Shinn; Paleoshorelines, Reefs, and a Rising Sea: South Florida, U.S.A. *Journal of Coastal Research* Vol. 7, No. 1 (Winter, 1991), pp. 203-229.) The same was reported from clastic deltas by Törnqvist and Hijma in *Nature Geosciences* 2012, who give various other examples. Maybe some of these works could be cited here.

Response: As suggested by Reviewer #2, we have included these two references in the modified manuscript. As pointed out by Reviewer #2, these two articles demonstrate decadal to centennial punctuated sea level rise during the Holocene. Our study, for the first time, supports the occurrence of punctuated sea level rise records during the Uppermost Pleistocene (~21000-11,500 cal years BP) mid-deglaciation.

Comment 3: The title is still awkward; why is there colon and why do you need events? I would simply say “Coralgal Reef Morphology Records Evidence of Punctuated Sea-Level Rise during Last Deglaciation.”

Response: As suggested by Reviewer #2 the title has been modified to – “Coralgal Reef Morphology Records Evidence of Punctuated Sea-Level Rise during Last Mid-Deglaciation”. Mid is added to the title to stress upon the middle part of last deglaciation during which the common terrace levels developed (current depth of common terraces lie between - 62-75 mbsl).

Comment 4: The figures still could be improved: First, all the A B etc labels are too big and have too large of a white background that distract from the figure itself. Second, all the bathymetry coloring should be in the same range. Currently all banks in Figure 2 have a different depth range. Third, Schlager’s backstepping model could be eliminated from Figure 3 and the backstepping arrow could be placed on Fig. 3 D

Response: First - All the A, B labels have been modified. Second – The bathymetric coloring is kept with different scales to showcase the morphology and the terrace depths better. The banks are partially buried and their exposed thickness varies from 10 to 25 m. Having a common coloring scheme would therefore limit display of the different morphological features. Third - Schlager’s backstepping model has been eliminated.

Reviewer # 3

Comment 1: This manuscript presents new high-resolution bathymetric and seismic data of drowned reefs in the Gulf of Mexico south of Texas. These new maps indicate the existence of extensive terraces at modern depths between 102 and 60 meters below sea level (mbsl). These terraces suggest stabilization of sea level at intervals since the last glacial maximum, likely punctuated by periods of more rapid sea level rise that led to the cessation of coralgall reef growth. Such rapid jumps in sea level are consistent with our understanding of the nonlinearities in the deglaciation of continental ice sheets and associated meltwater pulses (e.g. MWP-1a, MWP-1b, etc), which have been demonstrated in other sea level records.

The high-resolution of these mapped terraces is a novel accomplishment, indicating that the connection between deglaciation and sea level rise may include finer scale MWP events that have not yet been resolved until now. However, the lack of direct chronological constraints on these terraces makes the connection with deglacial events challenging. This is a revised manuscript, and the authors have taken steps to improve their estimates of paleo sea-level by converting the modern depths to a eustatic component, following with a comparison with the Lambeck et al. (2014) record to estimate the timing of sea level jumps. However, I think there are a number of points that need to clarification in the text, to determine what the authors have projected their data onto the Lambeck et al. curve and the NGRIP d18O record. In particular, I think the manuscript underestimates uncertainties in these projections, and I would like to see a more complete inclusion of these uncertainties. In the end, such uncertainties may preclude the connection of these terraces with particular climate events, limiting some of the conclusions of this paper.

Response: We appreciate that Reviewer#3 considers the high-resolution mapping of the terraces a novel accomplishment, which potentially indicates the presence of finer scale MWP events that as of yet have not been resolved. In the absence of direct chronological constraints, using the dates from previous studies, as well as the seismic data, the lifespan of the reefs is bracketed to ~8000 years, accreting up to 40- 50 m during early-mid part of the last deglaciation. Further, the high resolution bathymetry data in this study clearly demonstrate that four terrace levels from 62-75 meters are common to eight banks over a distance of 120 km, in addition to two deeper terrace levels at ~82 (identified via seismic data), and at ~94 mbsl for Harte bank (the deepest bank, as identified by multibeam bathymetry). This is therefore the only study where six terraces that developed during early-mid part of last deglaciation are identified. Rapid rise in sea level causes the reefs to backstep (Schlager, 2005), leading to development of terraces, and since the last glacial maxima the sea-level has been rising; therefore, understanding the nature of the sea-level rise since the last glacial maxima sheds light on the development of the terraces. The only high resolution sea level record to which these terraces could be compared is the NGRIP climate record. The comparison between these two records illustrates that four terraces correspond to interstadials, one to the stadial-interstadial transition, and one to a stadial. As mentioned by Reviewer#3 in later comments, the NGRIP is not a pure eustatic record, and so a perfect match of each terrace to unique warming events should not be expected. Additionally, the number of occurrences of warming events in the NGRIP record and the number of occurrences of terraces are found to be similar, further suggesting a linkage.

As suggested by Reviewer #3, additional uncertainties have been included in the revised manuscript (see Table 1, Figure 5B, and text). All assumptions, based on corrected terrace depths and ages, have been calculated, and are stated in the manuscript. In Fig 5B, the paleo terrace

depths (corrected terrace depths) with uncertainties are projected onto the Lambeck et al. (2014) record and their equivalent ages and uncertainties are estimated based on these projections. These ages, and associated uncertainties, are projected onto the NGRIP record. The age uncertainties do not preclude a connection between terrace occurrences and the warming events in the Greenland climate record; details are included as a response to later comments.

Comment 2 : Line 38 (and throughout) – Calibrated radiocarbon dates (calendar years) should include uncertainty. In addition, please indicate the calibration dataset you are using.

Response: The radiocarbon date calibration and their uncertainties were not included in the Droxler and Jorry (2013) and Belopolsky and Droxler (1999); therefore, we have conducted our own calibration analysis. Calib Rev 7.0.4 was used to calibrate the radiocarbon ages. Calibration dataset marina13.14c is used with $\Delta R = -30 \pm 9$. The new calibrated calendar year ages are, $11,901.5 \pm 335.5$ calendar years BP for the top of Dream Bank and 22361 ± 428 calendar years BP for the base of the Southern Bank. A new section is added in methods to display these calculations.

Comment 3: Line 41 – “This study further point...” should be “This study further points...”

Response: The manuscript has been modified.

Comment 4: Line 86-88 – Is there a particular range of depths in which the corallgal reefs can survive (photic ‘zone’)? The caption of Figure 3 groups 8 terraces that are within 3-4 depth of a

particular depth range. Is this coralgal reef habitat depth uncertainty? If so, shouldn't this be included as a source of uncertainty in your estimates of paleo sea level from these terraces?

Response: Reef building corals living in symbiosis with zooxanthallae cannot grow below the photic zone and require specific latitude, nutrient, temperature, and salinity conditions. In this study, we have demonstrated that, as in the Caribbean, *A. Palmata* and *A. Cervicornis*, the modern reef building corals growing to sea-level, most likely formed the reef framework of the south Texas shelf drowned reefs. The true coralgal nature of these reefs growing to sea level is also supported by the atoll and spur-groove morphologies observed in the high-resolution bathymetry data set (Fig 3).

Figure 3G displays depths to crest of ten banks, eight of which lie within a 3-4 m depth range from 57.5 – 61.8 mbsl(also described in the figure caption). This 3-4 m depth range does not represent terrace depth, but represents the top of the banks, and also indicates that the drowning of the banks likely occurred simultaneously.

Comment 5: Line 95 – You reference Fig. 4A. There is no Fig 4A. Do you mean Fig 3A?

Response: We meant Figure 5A to represent the hypsometric curves. The manuscript has been modified with this change.

Comment 6: Line 110 – First usage of GIA. Must define.

Response: The manuscript has been modified.

Comment 7: Line 114 – remove the word “simply”. Also, why is “keep up” in quotes?

Response: The manuscript has been modified.

Comment 8: Line 115 – comma before “because” is unnecessary

Response: The manuscript has been modified.

Comment 9: Line 118 – remove words “to be sure”

Response: The manuscript has been modified.

Comment 10: Line 121 – “Texas mud blanked” should be “Texas mud blanket”. Also, this is the second time you define TMB.

Response: The manuscript has been modified.

Comment 11: Line 132-133 – I do not understand this sentence. What is the word “become” referring to? Also, what is the “latter”?

Response: This sentence is meant to show that relative sea level incorporates the eustatic sea level signal and it is usually difficult to separate the two. However, if the drivers of

RSL rise are well constrained, then identifying the eustatic signal is possible. The sentence has been modified to make this point more clear.

Comment 12: Line 135 – define “GoM” if you intend to use it throughout.

Response: The manuscript has been modified

Comment 13: Line 136 – Need to include units of “years” after “21,000”

Response: The manuscript has been modified

Comment 14: Line 137-139. The process of estimating the GIA/subsidence corrections should be done iteratively. From what I understand in the text, the initial depth is linked with time using Lambeck et al. This time component is necessary to estimate the GIA/subsidence correction. But the eustatic/subsidence correction changes the implied projection onto time using Lambeck et al. Thus, a new estimate of time must be estimated from Lambeck to provide an appropriate estimate of GIA/subsidence. This is a major challenge for this manuscript. Since there is no direct chronology on the terrace, such an iterative projection of GIA/subsidence correction might lead to uncertainties. The authors should discuss their assumptions in this approach.

Also, in this projection of sea level onto time using Lambeck, are you including depth uncertainty in these projections? There should be uncertainty from measurements (most depths in Table 1 have uncertainty of +/- 1 to 1.5 m), coralgal reef habitat depth range, eustatic uncertainty

(from Lambeck et al.). These uncertainties must be included in terrace depth with time. Table 1 implies that the estimates of time for these terraces have no uncertainty. This is disingenuous.

Response: As per Reviewer#3's recommendation, the GIA/Subsidence corrections have been made iteratively. As advised by the Reviewer#3, a new estimate of time has been produced from Lambeck et al. (2014).

It was found that the GIA and subsidence vary less than two percent for the terrace depth zones, and three percent for the shallower terrace depth zones. Therefore, the new GIA and subsidence estimates have not been added to the modified manuscript. The assumptions and the new results have been added to the manuscript.

Yes, we include depth uncertainty into these projections. The uncertainty from depth measurements is defined by terrace zones. Indirect evidence shows *A. Palmata* or *A. Cervicornis* to be the main reef builder, but since the bathymetry maps display atoll and spurs-grooves morphologies, indicating that the reefs were growing at sea-level, the uncertainty due to coralgal reef habitat is assumed as zero. The 95% probability esl curve is used from Lambeck et al. (2014), so the uncertainties associated with eustacy are not considered.

Comment 15: Line 139 – “an indicator of RSL” – By including GIA and subsidence in your correction, aren't you providing an estimate of the eustatic sea level change that the terraces indicate?

Response: Reviewer#3 is correct. The manuscript has been modified.

Comment 16: Line 142-144 – This sentence is not clear. What is being projected onto what?

- Are you projecting terrace depths onto ice volume? Or are you projecting terrace depths onto NGRIP?
- Also, how does projecting anything onto NGRIP “identify the eustatic signal” as indicated in the first part of the sentence? NGRIP is not a pure eustatic signal, without some correction for $\delta^{18}\text{O}$ changes due to changes in temperature and precip.
- Are you projecting modern terrace depth onto NGRIP or eustatic corrected terrace depth onto NGRIP? Your table 1 makes it seem as though you are inferring ages using the raw terrace depth, not the eustatic corrected depth. This is not right.
- Are you including depth uncertainty in these projections?

Response: The sentence has been modified and made clearer. The paleo water depths (corrected depths) of the observed six common terrace levels identified on south Texas banks are projected onto the global eustatic sea-level curve and their equivalent ages with their uncertainties are estimated based on these projections. Then, these ages with associated uncertainties are projected onto the NGRIP $\delta^{18}\text{O}$ record.

NGRIP $\delta^{18}\text{O}$ record, as far as we know, is the only high-resolution climate record for the time period of the terrace occurrences. The comparison of NGRIP climate record with the high resolution terraces observed on south Texas banks is, therefore, the only possible opportunity to understand the cause and effect relationship between warm climate intervals, melting of glaciers (ice-stream/ ice sheet collapse), sea-level rise events, and terrace development. As observed in the modified Fig 5B, out of the six terraces, four terraces correspond to interstadial, one to the

stadial-interstadial transition, and one only to stadial. A perfect match of interstadials with terraces should not be expected, as mentioned by Reviewer#3.

Eustatic corrected depths are projected on the Fig 5B. Depth uncertainties are included in the projections, represented by terrace depth zones, defined on the basis of the minimum and maximum depths of a terrace. Uncertainty for each terrace depth zone is listed in Table 1. On the basis of the atoll and spurs-grooves morphologies, we conclude that the drowned reefs in this study were growing to sea level (as consistent with modern coralgall reefs). We assume, therefore, that the terraces were formed very near sea level.

Comment 17: Line 144 (and throughout) – the 18 in $\delta^{18}\text{O}$ should be superscript

Response: The manuscript has been modified.

Comment 18: Line 144-148. This statement suggests that the sea level jumps line up with warm periods (interstadials) in the NGRIP. There are a number of problems with this statement:

- As stated elsewhere in my comments, I think depth and chronological uncertainties are underestimated in this analysis. Therefore, the projection of sea level rise onto the NGRIP curve should be displayed as a band, NOT an individual line. Such uncertainty may cause this projection to span an interstadial and a neighboring stadial, thereby precluding the connection between the terraces and NGRIP warm periods.
- The NGRIP $\delta^{18}\text{O}$ curve is not solely temperature. This curve also includes source effects from precipitation. See LeGrande and Schmidt (2009, Climate of the Past)
- The two youngest projections (one at ~ 12.5 ka, another at ~ 12.9 ka) in Figure 5 appear to link

up with stadials (cool periods, not warm periods) in the NGRIP record.

Response: The statement Reviewer#3 is referring to has been modified. We agree that in the previous version the uncertainties were underestimated; as such, Table 1 has been modified in the revised manuscript. As suggested by Reviewer#3, the projection of time from the sea-level curve onto the NGRIP record is displayed as a band to take into account uncertainties (Fig 5B). In the updated figure, out of the six terraces, four terraces correspond to interstadial, one to the stadial-interstadial transition, and one to a stadial.

Comment 19: Line 148 – what is meant by “the cause and effect circle”. Be more specific.

Response: The manuscript has been modified to make the sentence clear.

Comment 20: Line 153 – what do you mean by “rapid”? Can you estimate the rate of sea level rise? Related to this question, what is the rate of sea level rise necessary to outpace coralgal reef growth.

Response: By rapid we define more than the current average rate of sea level rise of approximately 3 millimeters per year; the value during MWP 1A (the last deglaciation) reached up to 40 millimeters per year. The total reef thickness on the south Texas shelf is measured to be 40-50 m, developed between 21,000-11,000 Cal years BP, and therefore indicates an average rate of sea level rise of approximately 4-5 millimeters per year. The primary message of this manuscript is that sea level rise during last deglaciation was characterized by punctuated events,

and therefore the upper limit of sea level rise was greater than the long-term average of 4-5 millimeters per year.

The rates of sea level rise necessary to outpace corallgal reef growth vary (few mm's to few 10's of mm from one) location to other due to variations in water temperature, nutrient load, and salinity; these factors also play an important role in setting growth rates.

Comment 21: Line 154 – what is meant by a “non-steady process at a human time scale”?

Response: By non- steady, we indicate variation over time scales that are important to humans, so decades to centuries. This line has been modified.

Comment 22: Line 154-156. Awkward sentence. Revise.

Response: The manuscript has been modified.

Comment 23: Line 175. Should there be a hyphen in “sub sea floor”?

Response: Yes, a hyphen is added between sea-floor. The manuscript has been modified.

Comment 24: Figure Caption 2.

- Please define what is meant by “spurs” and “grooves”
- “well developed” should be hyphenated\

Response: The manuscript has been modified.

Comment 25: Figure Caption 3.

- “south-western margin of Baker Bank”. Do you mean “southeastern”?
 - It is strange to have a URL in the middle of a caption. Is this URL the citation for the photo?
- Also, what happens if the URL becomes inactive over time?

Response: The manuscript has been modified.

Comment 26: Figure 5A. What do the abbreviations “MB” and “CH” stand for in the lower right inset?

Response: MB stands for Multibeam and CH stands for Chirp Data. The abbreviations full definitions were located in the lower left corner. Since they were too small to read, their size has been increased.

REVIEWERS' COMMENTS:

Reviewer #3 (Remarks to the Author):

The revised manuscript is greatly improved and the authors have addressed my concerns regarding uncertainty and text clarity. In particular, I am glad to see the addition of Table 1, laying out these uncertainties in depths and chronological constraints.

The general finding that deglacial sea level rise was likely punctuated by abrupt sea level jumps is a novel result and relevant to our understanding of modern and future sea level rise. I support this manuscript for publication.

Reviewer #3 (Remarks to the Author)

The revised manuscript is greatly improved and the authors have addressed my concerns regarding uncertainty and text clarity. In particular, I am glad to see the addition of Table 1, laying out these uncertainties in depths and chronological constraints.

The general finding that deglacial sea level rise was likely punctuated by abrupt sea level jumps is a novel result and relevant to our understanding of modern and future sea level rise. I support this manuscript for publication.

Response: We have particularly paid attention to Reviewer # 3 comments, suggestions, and recommendations in modifying the manuscript in the text and figures, in particular in integrating detailed uncertainties in the estimates of the coralgal reef terraces corrected depths and their equivalent timing. We are glad that Reviewer #3 has recommended the manuscript for publication.